# Morphology and Molecular Phylogeny of the Genus *Stigeoclonium* (Chaetophorales, Chlorophyta) from China, Including Descriptions of the *Pseudostigeoclonium* gen. nov.

**DOI:** 10.3390/plants13050748

**Published:** 2024-03-06

**Authors:** Benwen Liu, Qiumei Lan, Qingyu Dai, Huan Zhu, Guoxiang Liu

**Affiliations:** 1Key Laboratory of Algal Biology, Institute of Hydrobiology, Chinese Academy of Sciences, Wuhan 430072, China; liubw@ihb.ac.cn (B.L.); lanqiumei23@mails.ucas.ac.cn (Q.L.); daiqingyu@ihb.ac.cn (Q.D.); huanzhu@ihb.ac.cn (H.Z.); 2University of Chinese Academy of Sciences, Beijing 100039, China

**Keywords:** *Caespitella*, Chaetophorales, morphology, molecular phylogeny, *Pseudostigeoclonium* gen. nov., *Stigeoclonium*

## Abstract

*Stigeoclonium* is a genus of green algae that is widely distributed in freshwater habitats around the world. The genus comprises species with variously developed prostrates and erect systems of uniseriate branched filaments and grows attached to a wide range of different surfaces. It holds significant promise for applications in water quality indicators, sewage treatment, and the development of high-value-added products. Nevertheless, our comprehension of *Stigeoclonium* remains unclear and perplexing, particularly regarding its fundamental systematic taxonomy. Recent molecular analyses have revealed that the morphologically well-defined genus *Stigeoclonium* is polyphyletic and requires taxonomic revision. Phylogenetic analysis based on a single molecular marker and limited samples is insufficient to address the polyphyletic nature of *Stigeoclonium*. In the present study, 34 out of 45 strains of *Stigeoclonium* were newly acquired from China. Alongside the morphological data, a concatenated dataset of three markers (18S rDNA + ITS2 + *tuf*A) was utilized to determine their molecular phylogeny. The phylogenetic analysis successfully resolved the broadly defined *Stigeoclonium* into three robustly supported clades (*Stigeoclonim tenue* clade, *S. farctum* clade, and *S. helveticum* clade). The morphological characteristics assessment results showed that the cell type of the main axis-producing branch, considered a crucial morphological characteristic of the *Stigeoclonium* taxonomy, did not accurately reflect the real phylogeny of the genus. A new taxonomical classification of the genus *Stigeoclonium* was proposed based on zoospores’ germination types, which aligned well with the phylogenetic topologies. Species where zoospores showed erect germination (*S. helveticum* clade) formed a distinct monophyletic clade, clearly separated from the other two clades, with zoospores showing prostrate germination or pseudo-erect germination. Consequently, a new genus, *Pseudostigeoclonium* gen. nov., is suggested to include all species in the broadly defined *Stigeoclonium* with zoospores with erect germination. The taxonomic diversity is supported by distinctive morphological differences and phylogenetic divergence within the broadly defined *Stigeoclonium* identified in this study. Further evaluation of the genus *Stigeoclonium* is necessary, especially via examining additional specimens and re-evaluating morphological characters under precisely defined laboratory conditions.

## 1. Introduction

*Stigeoclonium* is an important primary producer that is widely distributed in freshwater ecosystems around the world. They often attach to inorganic substrates or other large algae and higher aquatic plant surfaces. Some species of *Stigeoclonium* have high application prospects in terms of water quality indication, wastewater treatment [1,2,3], and lipid production [4].

The genus *Stigeoclonium*, comprising approximately 80 described species, was initially established by Kützing [5] and belongs to the order Chaetophorales (Chlorophyceae, Chlorophyta) [6]. *Stigeoclonium* was originally defined based on the type species *Stigeoclonium tenue*, encompassing algae forming vibrant green tufts with uniseriate branched heterotrichous filaments [5]. Further delimitation of the genus occurred through the examination of various vegetative traits, such as cellular dimensions, branching degree, the presence/absence of hairs, thallus color, and habitat, resulting in the description of 29 species [7]. Subsequent taxonomic revisions by different phycologists led to varied concepts and proposed species. Islam [8] critically reviewed the genus based on 73 specimens worldwide, recognizing 28 species, with only 10 being considered valid. Printz [9] reported 42 species, but 11 remained not fully understood. These phycologists primarily delimited *Stigeoclonium* species based on branching types and the shape and size of cells in the erect system.

*Stigeoclonium* is a heterotrichous alga, which is composed of an erect system and a prostrate system. Cox and Bold [10] introduced a species concept, focusing on the morphological features of the prostrate system and considering it more stable than the erect system. This approach identified seven species from 81 *Stigeoclonium*-like specimens collected in Texas, which belonged to three groups. Francke and Simons [11] refined these methods, confirming four groups (*S. helveticum* group, *S. aestivale* group, *S. tenue* group, and *S. farctum* group) from 150 strains collected in the Netherlands. Simons et al. [12] further merged groups based on detailed studies on the morphological development of the prostrate system of *Stigeoclonium*, with particular emphasis on the types of zoospore germination, resulting in three groups: *S. helveticum*, *S. tenue*, and *S. farctum*, each with a single species.

Despite these efforts, universally recognized taxonomic criteria for *Stigeoclonium* remain elusive. Various authors have proposed species and variants based on different concepts. John et al. [13] focused on the morphology of the erect system, describing nine species. Branco et al. [14] identified six species in southeastern Brazil by considering the characteristics of both prostrate and erect systems. Skinner and Entwisle [15] described three species in Australia following the species concept of Simons et al. [12].

The genus *Stigeoclonium* has undergone comprehensive study for nearly two centuries, resulting in numerous species descriptions based on morphology. However, the traditional taxonomic framework established by Kützing [5] has persisted without significant changes. The genus *Stigeoclonium* was considered a well-defined monophyletic genus for a long time until Caisová et al. [16] first used SSU rDNA to study the phylogeny of *Stigeoclonium*, which contained nine molecular sequences of six species of *Stigeoclonium*, confirming that *Stigeoclonium* was polyphyletic, challenging the previous notion. Notably, species sharing the same name, including the type species *S. tenue*, did not form cohesive clusters, adding complexity to the phylogenetic understanding of *Stigeoclonium* [17,18].

While molecular phylogenetic studies have primarily concentrated on the order Chaetophorales as a whole, investigations specific to *Stigeoclonium* have been limited. To address these challenges, the present study generated new sequences for 18S rDNA, ITS2, and *tuf*A. A concatenated dataset, combined with morphological data, was employed to (1) re-evaluate morphological characteristics and (2) reassess the broadly defined genus *Stigeoclonium*, leading to the proposal of a new genus.

## 2. Results

### 2.1. Taxonomic Implications

*Pseudostigeoclonium* B. Wen Liu, Q. Mei Lan, Q. Yu Dai, Huan Zhu et G. Xiang Liu gen. nov.

Diagnosis: Turf-like thallus; cushion-forming; 1–5 cm. Filaments consist of erect and prostrate systems. The erect filaments are unbranched in the actively growing stage or sparsely branched in the older stage. The branches usually alternate, are rarely approximate or opposite, and branches are long, bluntly attenuating, or ending in a colorless hair. Cells of the main filament and branches are cylindrical and either slightly constricted or not. Chloroplasts are single and parietal; pyrenoid 1–3; typically two per cell. The prostrate systems are underdeveloped, attached to the substrate by a sparse, rhizoid-like basal system developed from the lower or middle cells. Sexual reproduction produces biflagellate isogametes, and asexual reproduction produces four-flagellated zoospores. Zoospore germination is erect. The first cross-wall of the attached germling is parallel to the substrate.

Etymology: The genus is named for its morphological similarity to the genus *Stigeoclonium*.

Type species (designated herein): *Pseudostigeoclonium helveticum* (Vischer) B. Wen Liu, Q. Yu Dai, Huan Zhu et G. Xiang Liu comb. nov.

*Pseudostigeoclonium helveticum* (Vischer) B. Wen Liu, Q. Mei Lan, Q. Yu Dai, Huan Zhu et G. Xiang Liu comb. nov.

Synonym: *Stigeoclonium helveticum* Vischer.

Formaldehyde-fixed material of *Pseudostigeoclonium helveticum* in this study was deposited in the Freshwater Algal Herbarium (HBI), Institute of Hydrobiology, Chinese Academy of Sciences, Wuhan, Hubei Province, China, as specimen no. WDLC201608.

Locality: Heilongjiang province, China; on a rock in a stream; freshwater.

### 2.2. Morphological Observations

Species of *Stigeoclonium* were highly variable in morphology, especially in the erect system. For instance, *Stigeoclonium farctum* specimens displayed different morphologies when growing epiphytically on a plastic cup in their natural habitat compared with those cultured in medium. Filaments cultured on solid medium were abundantly branched, whereas those in liquid medium had extremely long, upright filaments with rare branching (Figure 1).

In this study, three types of zoospore germination were clearly observed (Figure 2). The genus *Stigeoclonium* was grouped into three clades based on zoospore germination types:

The *Stigeoclonim helveticum* clade (*Pseudostigeoclonium* gen. nov.): The *Stigeoclonim helveticum* clade exclusively included the species *Stigeoclonium*, primarily including *Stigeoclonium helveticum* and *Stigeoclonium* sp. This clade was characterized by the strictly erect germination of zoospores (Figure 2A) and a prostrate system consisting of one *Ulothrix*-like holdfast cell or some very short prostrate filaments and/or slender rhizoids growing out from the basal cell of an erect axis [19,20].

The *Stigeoclonim tenue* clade: The *Stigeoclonim tenue* clade included some species of *Stigeoclonium* and the genus *Caespitella*. This clade was characterized by an approximately prostrate germination of zoospores. The original zoospore cell unipolar germination usually gave rise to a lateral outgrowth at one side to first form the prostrate part, and later, a pointed erect filament began to develop (Figure 2D). We placed all species with an approximately prostrate germination of zoospores and an irregular and mostly extensive open prostrate system, with or without rhizoids, into this clade.

The *Stigeoclonim farctum* clade: The *Stigeoclonim farctum* clade comprised some species of *Stigeoclonium* and the genus *Fritschiella*. This clade was characterized by an approximately prostrate or pseudo-erect germination of zoospores (Figure 2B,C,E). What was different from the *Stigeoclonim tenue* clade was that the original zoospore cell bipolar germination gave rise to a lateral outgrowth on two sides to form the prostrate part, with the exception of the *Fritschiella tuberosa* (approximately gregarious on moist soil or silt, including drying rainwater puddles), which showed the original unipolar germination of zoospore cells, and *S. aestivale,* which showed the original unipolar germination of zoospore cells, to first form the erect part. We placed all species with an approximately prostrate and pseudo-erect germination of zoospores, a regular compact development of the prostrate system that results in a closed, pseudoparenchymatous disk, an open star-like form, or intermediate forms and a poorly developed erect system, into this clade.

In addition, three cell types of the main axis-producing branch were observed (Figure 3). Species with a main axis and primary branches were similar, and usually, no specialized or modified cells were present on the main axis producing the branches (Figure 3A). Species showed transitions in which some differentiation began between the cells of the main axis. Some cells gradually became long and some short, with the latter commonly producing branches (Figure 3B). Species with a main axis usually consisted of two types of cells. The long cells usually did not produce branches, while the small and short cells usually produced lateral primary branches (Figure 3C).

### 2.3. Phylogenetic Analysis of the Stigeoclonium

A total of 135 new sequences, including 18S rDNA, ITS2, and *tuf*A, were generated in this study. The best models were selected for the BI analysis: TrNef + I + G for 18S rDNA, GTR + G for ITS2 rDNA, GTR + G for the first codon position matrix of *tuf*A, and GTR + I + G for the second and third codon positions matrix. The GTRGAMMA model was used for maximum likelihood analysis.

A saturation index analysis (*Iss* = 0.386 < *Iss.c* = 0.802) showed that the sequence matrix was not saturated and could be used for phylogenetic analysis. The 71-taxa alignment consisted of 2687 positions (18S rDNA + ITS2 + *tuf*A) (see Appendix B). A total of 743 sites in these nucleotides were variable, of which 573 sites were parsimoniously informative and 170 sites were singleton sites. The average amount of A, T, C, and G was 24.69%, 26.04%, 20.89%, and 28.38%, respectively, of which A + T (50.73%) was greater than G + C (49.27%). The transition/transversion ratio was 1.61.

The phylogenetic trees created using the Bayesian and ML methods showed similar topologies to previous studies [21,22]. The monophyly of the Chaetophorales was strongly supported, including six widely accepted families: Schizomeridaceae, Aphanochaetaceae, Barrancaceae, Uronemataceaea, Fritschiellaceae, and Chaetophoraceae [19] (Figure 4).

*Stigeoclonium* species were dispersed over the family Fritschiellaceae or family Chaetophoraceae and intermixed with the other genera. The family Chaetophoraceae and family Fritschiellaceae diverged into three well-supported sister lineages. The family Chaetophoraceae included the genera *Chaetophora*, *Draparnaldia,* and the *Stigeoclonium helveticum* clade. The *S. helveticum* clade showed basal divergence as a sister of the *Chaetophora* and *Draparnaldia*, while the family Fritschiellaceae comprised the genus *Chaetophoropsis*, *S. tenue* clade, and *S. farctum* clade. The *S. tenue* clade showed basal divergence as a sister of the *Chaetophoropsis* and *S. farctum* clade. The *S. tenue* clade and *S. farctum* clade, with a similar morphology, were well separated from each other. The three clades of *Stigeoclonium* were further split into seven molecular groups. All *Stigeoclonium* species were dispersed over these two families and were recovered as independent monophyletic clades (*Stigeoclonium helveticum* clade, *S. tenue* clade, and *S. farctum* clade) with moderate to robust support values (BP/PP, 80/0.85 to 100/1.00). Species of *Stigeoclonium* with zoospore germination of an erect type (*Stigeoclonium helveticum* clade) formed a monophyletic clade in the Chaetophoraceae, with a robust support value (BP/PP, 100/1.00), which obviously branched independently of the *S. tenue* clade and *S. farctum* clade (within the Fritschiellaceae) with zoospore germination prostrate or the pseudo-erect type in our phylogenetic trees. In addition, multiple sequences with the same species name were phylogenetically distantly related, including the type species of the genus *Stigeoclonium* (*S. tenue* in bold; Figure 1) and *Caespitella pascheri* (type species of *Caespitella*). Two sequences with the same name, *S. tenue*, were dispersed over the Fritschiellaceae and the Chaetophoraceae instead of clustering together.

## 3. Discussion

Based on concatenated datasets and additional samples, we still had to accept the fact that the *Stigeoclonium* was paraphyletic. Clearly, the morphologically defined genus *Stigeoclonium* was unnatural and needed re-evaluation. This study obtained a well-resolved molecular phylogeny with moderate to high support values for *Stigeoclonium*. All species of *Stigeoclonium* were divided into three clades with robust support, located in either the family Fritschiellaceae or the family Chaetophoraceae. This provided a basis for further discussion on the phylogenetic relationships within the paraphyletic genus *Stigeoclonium*.

In cases where paraphyly or polyphyly is demonstrated within a genus, a common approach is to designate one clade as “genus sensu stricto” and analyze the morphological characteristics used in the original diagnosis [16]. Before that, it is necessary for us to re-examine its type species. However, re-examination of the type species revealed two stains of *S. tenue* (*S. tenue* CCAC 3492B HF920647 and CCAP 477/11A FN824374), which are clearly unrelated and located in Fritschiellaceae and Chaetophoraceae, respectively. Based on a comprehensive sample observation and literature analysis, the zoospore germination type of the type species (*S. tenue*) was consistently identified as prostrate [10,11,12]. On the other hand, recent research has provided evidence indicating that taxa within the family Fritschiellaceae exhibit zoospore germination of the prostrate or pseudo-erect type and, in contrast, taxa belonging to Chaetophoraceae display zoospore germination of the erect type [19]. Consequently, the inference that was drawn was that *S. tenue* CCAC 3492B (HF920647), situated in the Fritschiellaceae, was more likely to represent *Stigeoclonium* sensu stricto. This conclusion aligned with the understanding that the germination characteristics of zoospores played a crucial role in delineating the taxonomy of *Stigeoclonium*.

After identifying *Stigeoclonium* sensu stricto, we next tried to find some effective morphological characteristics to revise the genus *Stigeoclonium*. The characteristic cell type of the main axis-producing branch was considered the most important morphological characteristic in the traditional taxonomy of *Stigeoclonium* by Islam [8], although it did not reflect the real phylogeny of the genus *Stigeoclonium* in this study. Instead, the characteristics of zoospore germination types correlated well with the phylogenetic results. Drawing upon the morphological features of the prostrate system and the germination type of zoospores, various phycologists have categorized the genus *Stigeoclonium* into distinct groups [10,11,12]. Despite the fact that none of these grouping schemes matched the phylogenetic results well, the important studies mentioned above [10,11,12] provide a valuable indication that the germination type of zoospores might serve as a dependable morphological characteristic [20]. This observation is consistent with findings from prior studies [19], reinforcing the significance of zoospore germination type in understanding the taxonomy of *Stigeoclonium*.

The broadly defined genus *Stigeoclonium* diverged into three well-supported clades in our phylogenetic analyses using a concatenated dataset, which was initially unexpected. By observing the zoospore germination type of representative species and combining the phylogeny results, the genus *Stigeoclonium* could be divided into at least six groups—the *S. helveticum* group, *S. aestivale* group, *S. variabile* group, *S. farctum* group, *S. tenue* group and *S. pascheri* group—and three clades—the *S. tenue* clade, *S. farctum* clade, and *S. helveticum* clade. The life history and reproductive development characteristics of *S. helveticum* (*S. helveticum* group) have been well studied, providing us with many taxonomic insights [23]. All species with erect zoospore germination types (*S. helveticum* group) of the broadly defined *Stigeoclonium* formed a separate, monophyletic clade that was clearly separate from the *S. tenue* clade (*Stigeoclonium* sensu stricto) and the *S. farctum* clade with zoospore germination of the prostrate and pseudo-erect types. Thus, in light of the morphological and molecular differences, we erected a new genus, *Pseudostigeoclonium* gen. nov., and amended the *Stigeoclonium*. The genus *Pseudostigeoclonium* was newly erected due to its unique morphological characters, as follows: the erect filaments are unbranched in the actively growing stage or sparsely branched in older stages; the prostrate systems are underdeveloped, attached to the substrate by a sparse, rhizoid-like basal system developed from the lower or middle cells; zoospore germination is of the erect type. This clearly distinguishes *Pseudostigeoclonium* from *Stigeoclonium* sensu stricto.

Usually, samples with typical characteristics of *Stigeoclonium* are confidently classified within the genus. However, some species of *Stigeoclonium* (such as *S. terrestre* and *S. pascheri*) spun off to create the new genera [24,25]. Vischer [25] discovered and isolated a new *Stigeoclonium*-like taxon with cell division that was entirely apical, distinguishing it from the *Stigeoclonium* with intercalary cell division and, rather than describing it as a new species of *Stigeoclonium*, created the new genus *Caespitella* with the type species *C. pascheri.* There is a lack of consensus among phycologists regarding this taxonomic approach. Printz [9] and Bourrelly [26] recognized this genus based on the morphological characteristics of the erect system. However, Cox and Bold [10] reincorporated the genus into *Stigeoclonium* based on detailed life history observations, which was accepted by Shyam and Sarma [27]. Caisová et al. [16] endorsed the establishment of the genus *Caespitella* based on the limited samples and molecular data of *Stigeoclonium*. In this study, two stains of *Caespitella pascheri* (*S. pascheri* HB201611 OP236750 and SAG 410-1 FN824387) (type species of *Caespitella*) were not independent of the genus *Stigeoclonium*; instead, they were dispersed within the *S. pascheri* group. Moreover, the key morphological characteristics (zoospore germination types and prostrate system) were found to be more reliable and were not significantly different between *Caespitella pascheri* and other *Stigeoclonium*. Thus, proposing *Caespitella pascheri* as a separate genus raised questions. Vischer [25] was, at the time, intensively studying both *S. helveticum* and *C. pascheri*. These two species exhibited significant differences in growth pattern, both in liquid and solid agar, which might have influenced Vischer’s belief that they belonged to different genera. In fact, apical cell division and intercalary cell division have been observed in the same species [10]. Apical cell division is common in prostrate filaments, while intercalary cell division is common in erect filaments [10]. The authentic strains of *C. pascheri* and *S. helveticum* could also be well distinguished on the phylogenetic tree. Considering the crucial evidence presented by Cox and Bold [10] and the molecular phylogeny in this study, it is suggested that *C. pascheri* should be considered a member of *Stigeoclonium*.

It is noteworthy that the sequences with the same species name were not clustered into one clade in the genus *Stigeoclonium* (e.g., *S. tenue*, *S. pascheri*), which may be for the following reasons: (1) the presence of cryptic species within the genus *Stigeoclonium*, where species share similar morphological characteristics but exhibit significant differences in molecular data; (2) insufficient knowledge among phycologists regarding the genus *Stigeoclonium*, leading to the use of inadequate morphological characteristics for classification and resulting in the definition of multiple species as a single species; (3) the prevalence of morphological plasticity in *Stigeoclonium* species.

It is also essential to emphasize that the precise identification of most species is challenging, and several specimens exhibit characteristics that are significantly different from known species in the genus *Stigeoclonium*, as reported in previous studies [8,9,28]. Consequently, we refrain from drawing conclusions or hastily proposing new species, primarily due to the limited specimens and the phenotypic plasticity [11,29,30].

Currently, reliable morphological characteristics that can be used to distinguish between *Stigeoclonium* sensu stricto and the *S. farctum* clade still have not been identified. Although the zoospore germination types and the prostrate system are important morphological features, they appear to be unsuitable for differentiating between the mentioned clades and species within *Stigeoclonium*. In the future, with an ample supply of specimens, a well-resolved molecular phylogeny, as presented in this study, could be used to re-evaluate morphological characteristics in *Stigeoclonium* under precisely defined laboratory conditions.

## 4. Materials and Methods

### 4.1. Taxon Sampling and Culture Conditions

A total of 45 Chaetophorales specimens, including 34 *Stigeoclonium* specimens, were collected from China. Voucher specimens were deposited in the Freshwater Algal Herbarium, Institute of Hydrobiology, Chinese Academy of Sciences (HBI). The sample collection information and GenBank accession numbers are listed in Appendix A.

Each sample was preserved in either 10% formalin or viviality for morphological observation and in 100% ethanol for DNA extraction. Natural samples were isolated using forceps and dissecting needles under an Olympus SZX7 microscope (Olympus Corp., Tokyo, Japan), rinsed with double-distilled H_2_O, and cultured in culture dishes on sterilized BBM medium [31] solidified with 1.2% agar, at 20 °C, under the photon fluence rate of 15–35 μmol m^−2^ s^−1^, in a 14/10 h light–dark cycle, and transferred into fresh solid medium every week. After approximately 48 h of growth at 20 °C in the dark to induce zoospore liberation, microphotographs were taken using an Olympus BX53 light microscope (Olympus Corp., Tokyo, Japan) with the differential interference contrast method under an oil immersion objective lens.

### 4.2. DNA Extraction, PCR Amplification, and Sequencing

Genomic DNA was extracted using an Axygen DNeasy Plant Kit (Axygen Biotechnology, Hangzhou, China), according to the manufacturer’s instructions. Approximately 15 mg of filaments was added to 1 mL of 0.5 mm glass beads and 350 μL of phosphate buffer solution (PBS, pH 7.0), followed by bead beating at 4800 rpm for 2 min in a mini-beadbeater (Model 3110BX, Biospec Products, Bartlesville, OK, USA). The polymerase chain reaction (PCR) of the 18S rDNA was amplified according to Medlin et al. [32]. The sequence amplification profile consisted of an initial 5 min denaturing step at 95 °C, 34 cycles of denaturing at 94 °C for 45 s, 30 s annealing at 55 °C, 90 s extension at 72 °C, and a final extension of 10 min at 72 °C. The primer and amplification procedures of ITS sequence followed those of Hayakawa et al. [33], and *tuf*A followed those of Famà et al. [34]. ContigExpress Project (Invitrogen, Grand Island, New York, NY, USA) was used to edit low-quality regions and assemble the partial sequences. Some excised PCR products were cloned into a pMD18-T vector and transferred into DH5α *E. coli* competent cells (Takara Bio Inc., Otsu, Shiga, Japan). The universal sequencing primers were M13F and M13R [35]. All samples were then sent to WuHan Tsingke BioTech Co., Ltd., Wuhan, China, for sequencing.

### 4.3. Phylogenetic Analyses

Additional 18S rDNA, ITS2, and *tuf*A sequences of Chaetophorales were downloaded from GenBank (http://www.ncbi.nlm.nih.gov/ (accessed on 19 January 2024)) for analyses. For the analyses performed in this study, a 71-taxa alignment with a concatenated dataset of three markers (18S rDNA + ITS2 + *tuf*A) was generated.

Each gene set was initially aligned with MAFFT 7.2 [36] and manually refined using Seaview v. 4.32 [37]. jModelTest v.2.1.4 [38] was used to select the best-fitting evolutionary models for each marker according to the Bayesian information criterion calculations. All genes were then concatenated using Phyutility 2.2 [39] (see Appendix B). Mutational saturation was evaluated by looking at variable positions of the alignments using DAMBE 5.6 [40].

Phylogenetic reconstructions were performed using RAxML 8.0 [41] and MrBayes3.1.2 [42]. Bootstrap analyses with 1000 replicates of the ML dataset were performed to estimate the statistical reliability. For Bayesian analyses, Markov chain Monte Carlo analyses were run with four Markov chains (three heated; one cold) for 4 × 10^6^ generations, with trees sampled every 1000 generations. It was assumed that a stationary distribution was reached when the average standard deviation of the split frequencies between two runs was lower than 0.01. The first 25% of the calculated trees were discarded as burn in, and the remaining samples were used to construct a Bayesian consensus tree and to infer posterior probabilities. The bootstrap values and posterior probabilities are presented at the nodes. The resulting phylogenetic trees were edited using Figtree 1.4.2 (http://tree.bio.ed.ac.uk/software/figtree/ (accessed on 19 January 2024).

## Figures and Tables

**Figure 1 plants-13-00748-f001:**
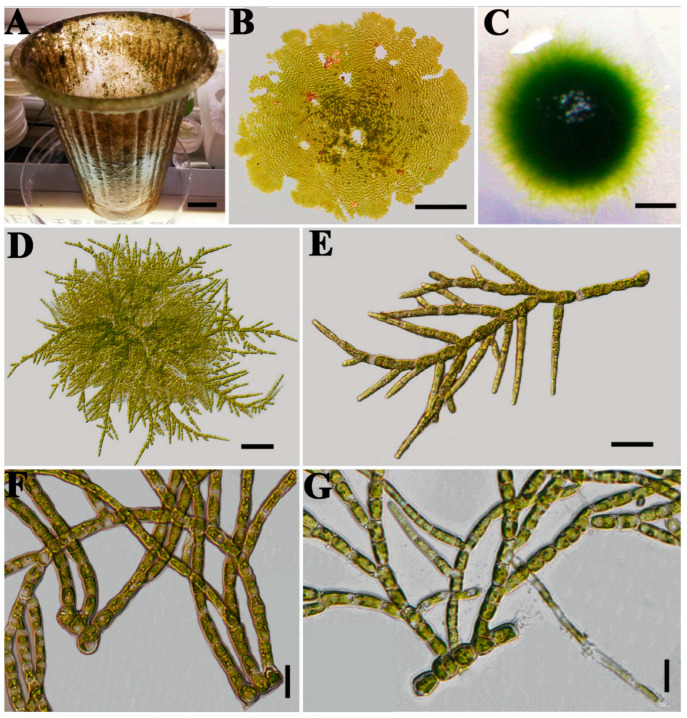
*Stigeoclonium farctum*: (**A**,**B**) growing on a plastic cup in a natural environment; (**C**–**E**) growing on a solid medium; (**F**,**G**) showing part of the erect and prostrate system in a liquid medium. Scale bar: A = 1 cm, B = 50 μm, C = 500 μm, D = 50 μm, E = 20 μm, F = 50 μm, G = 20 μm.

**Figure 2 plants-13-00748-f002:**
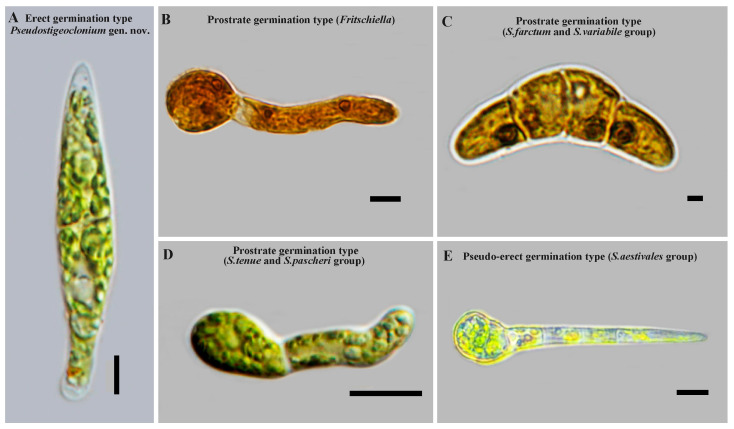
Zoospore germination types of the genus *Stigeoclonium*. Scale bar: (**A**) 5 μm, (**B**) 5 μm, (**C**) 2 μm, (**D**) 10 μm, (**E**) 5 μm.

**Figure 3 plants-13-00748-f003:**
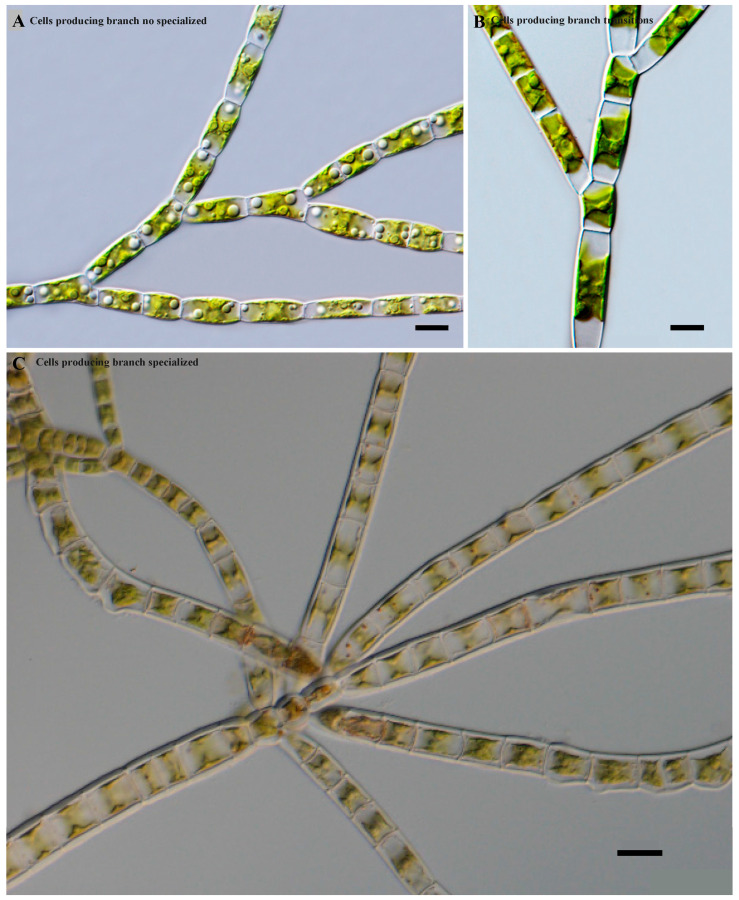
Cell types of the main axis-producing branch. Scale bar: (**A**) 10 μm, (**B**) 10 μm, (**C**) 20 μm.

**Figure 4 plants-13-00748-f004:**
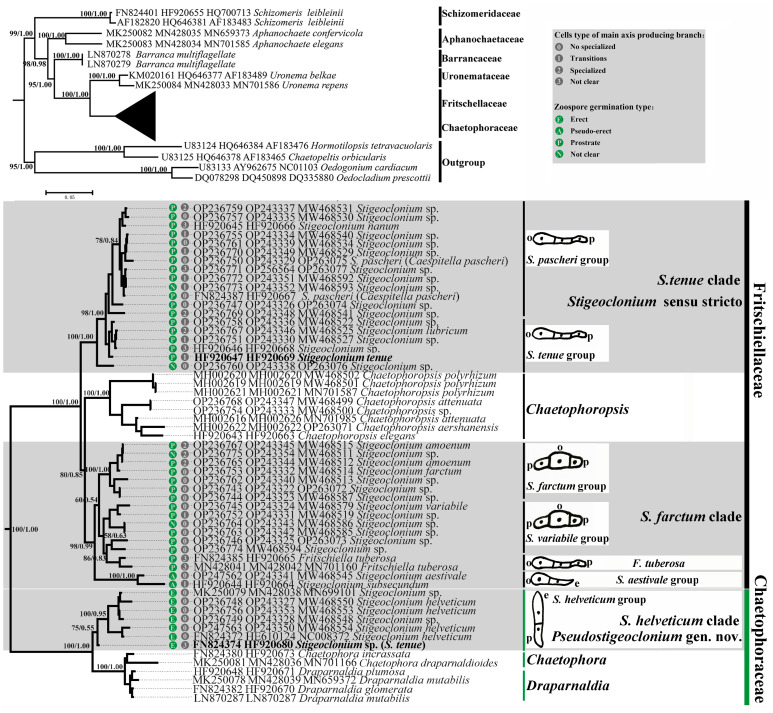
ML and Bayesian phylogenetic tree constructed using a concatenated dataset (18S rDNA + ITS2 + *tuf*A) of the Chaetophorales. The numbers on the nodes represent the bootstrap support values (BP)/posterior probabilities (PP) above 50/0.50. Two sequences of *S. tenue* from our study are shaded in bold.

## Data Availability

Data are contained within the article and its Appendix A.

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
