# Peer review of "Morphology and Molecular Phylogeny of the Genus Stigeoclonium (Chaetophorales, Chlorophyta) from China, Including Descriptions of the Pseudostigeoclonium gen. nov."

_plants, 2024, doi:10.3390/plants13050748_

Round 1
Reviewer 1 Report
Comments and Suggestions for Authors
Technical coments:
line 12 - if the alga grows epiphytically, as it is written, it is not on different substrates but only on plants. Suggestion - "grows attached"
line 57 - alga, not algae
line 281 - year of publication missing after Vischer
lines 285-286: apical cell division and intercalary cell division is more correct expression than "cell division apical... and cell division intercalary..."
Comments on the taxonomic description:
The paper is oriented towars a really variable and problematic genus Stigeoclonium and provides a novel approach to the taxonomy of this genus - based on combination of zoospore germination and well-done molecular-genetic studies. In addtion, a new genus - Pseudosigeoclonium - was proposed. However, the features used and decrived in the descriptions of Stigeoclonium and Pseudostigeoclonium are not sufficient enough and do not cover all the diagnostic features of green algae. Nothing is written about the shape, position and number of plastids and of the pyrenoids in the cells of main axis and smaller cells, and the reproduction by zoospores only is used without discussion of akinetes and sexual reproduction (isogamy?). Such information has to be added for the completeness of the taxonomic description and has to be easily available from the observed cultures, described in the paper.
Author Response
Reviewer: 1
- line 12 - if the alga grows epiphytically, as it is written, it is not on different substrates but only on plants. Suggestion - "grows attached"
Re: P1L12. Thank you for your comments. We have fixed them.
- line 57 - alga, not algae
Re: P2L57. Thank you for your comments. We have fixed them.
- line 281 - year of publication missing after Vischer
Re: P8L281. Thank you for your comments. I am sorry for such mistakes. We have fixed them.
- lines 285-286: apical cell division and intercalary cell division is more correct expression than "cell division apical... and cell division intercalary..."
Re: P9L285-287. Thank you for your comments. We highly agree with you. We have fixed them.
- Comments on the taxonomic description:
The paper is oriented towars a really variable and problematic genus Stigeoclonium and provides a novel approach to the taxonomy of this genus - based on combination of zoospore germination and well-done molecular-genetic studies. In addtion, a new genus - Pseudosigeoclonium - was proposed. However, the features used and decrived in the descriptions of Stigeoclonium and Pseudostigeoclonium are not sufficient enough and do not cover all the diagnostic features of green algae. Nothing is written about the shape, position and number of plastids and of the pyrenoids in the cells of main axis and smaller cells, and the reproduction by zoospores only is used without discussion of akinetes and sexual reproduction (isogamy?). Such information has to be added for the completeness of the taxonomic description and has to be easily available from the observed cultures, described in the paper.
Re: P1L17-18. Thank you for your comments. We highly agree with you. We have added relevant information:
Diagnosis: Turf-like thallus; cushion-forming; 1–5 cm. Filaments consist of erect and prostrate systems. The erect filaments are unbranched in the actively growing stage or sparsely branched in the older stage. The branches usually alternate, are rarely approxi-mate or opposite, and branches are long, bluntly attenuating, or ending in a colorless hair. Cells of the main filament and branches are cylindrical, and are either slightly constricted or not. Chloroplasts are single and parietal; pyrenoid 1–3; typically two per cell. The prostrate systems are underdeveloped, attached to the substrate by a sparse, rhizoid-like basal system developed from the lower or middle cells. Sexual reproduction produces biflagel-late iso-gametes, and asexual reproduction produces four-flagellated zoospores. Zoospore germination is erect. The first cross-wall of the attached germling is parallel to the substrate.
Reviewer 2 Report
Comments and Suggestions for Authors
The authors describe a new genus of chaetophoralean green algae based on developmental and molecular criteria. The paper is well illustrated (the photographs are particularly clear); however, the phylogenetic tree (Fig. 4) needs to be enlarged so that the fonts are not microscopic. The use of English needs considerable editorial attention by a native English speaker. Overall, I find that this is a useful contribution to the taxonomy of Chaetophorales and is worthy of publication.
My major complaint about the current paper is that the authors to not give sufficient credit to previous authors who examined taxonomy and development of Stigeoclonium and used spore germination patterns as key features (i.e., Michetti et al. 2010). The paper by Simons & van Beem (1986. Phycologia 26: 356-362) discusses the life history and reproductive development of S. helvetica. The unique features of the life history should have made it into the discussion of the new genus. Are the reproductive features useful as part of the diagnosis of the new genus?
The authors need to cite this paper:
Karina M. Michetti, Patricia I. Leonardi, Eduardo J. Cáceres. 2010. Morphology, cytology and taxonomic remarks of four species of Stigeoclonium (Chaetophorales, Chlorophyceae) from Argentina. Phycological Research 58(1): 35-43.
https://doi.org/10.1111/j.1440-1835.2009.00556.
Differences in spore germination patterns in key species are described. This paper also offers differences in chromosome numbers for each species.
Comments on the Quality of English LanguageMy apologies for not having time to undertake the editorial work necessary to bring the English up to an international standard.
Author Response
Reviewer: 2
- The authors describe a new genus of chaetophoralean green algae based on developmental and molecular criteria. The paper is well illustrated (the photographs are particularly clear); however, the phylogenetic tree (Fig. 4) needs to be enlarged so that the fonts are not microscopic. The use of English needs considerable editorial attention by a native English speaker. Overall, I find that this is a useful contribution to the taxonomy of Chaetophorales and is worthy of publication.
Re: Thank you for your comments. Sorry for the unclear display of the Figures. We have tried our best to modify it and hope to get your approval. Please see the Figure 4. Moreover, native English speakers have carefully revised the language of our MS. Please see the file English-Editing-Certificate-77600.pdf
- My major complaint about the current paper is that the authors to not give sufficient credit to previous authors who examined taxonomy and development of Stigeoclonium and used spore germination patterns as key features (i.e., Michetti et al. 2010). The paper by Simons & van Beem (1986. Phycologia 26: 356-362) discusses the life history and reproductive development of S. helvetica. The unique features of the life history should have made it into the discussion of the new genus. Are the reproductive features useful as part of the diagnosis of the new genus?
Re: Thank you for your comments. Sorry for the insufficient discussion and citation of previous research works. We have fixed them.
‘Despite the fact that none of these grouping schemes matched the phylogenetic results well, they provide a valuable indication that the germination type of zoospores might serve as a dependable morphological characteristic (Michetti et al. 2010)’.
‘The life history and reproductive development characteristics of S. helveticum (S. helveticum group) have been well studied and given us many taxonomic insights (Simons & van Beem, 1987)’.
According to literature reports, the Stigeoclonium has the variety of complex life history modes, but in this study only asexual reproduction through zoospores was observed. Whether reproductive traits are useful is not known but is worthy of attention, especially for Stigeoclonium species with great morphological plasticity.
- The authors need to cite this paper: Karina M. Michetti, Patricia I. Leonardi, Eduardo J. Cáceres. 2010. Morphology, cytology and taxonomic remarks of four species of Stigeoclonium (Chaetophorales, Chlorophyceae) from Argentina. Phycological Research 58(1): 35-43. https://doi.org/10.1111/j.1440-1835.2009.00556. Differences in spore germination patterns in key species are described. This paper also offers differences in chromosome numbers for each species.
Re: Thank you for your suggestions. We highly agree with you. We have cited this reference. Indeed, this reference has given us a good enlightenment.
‘Despite the fact that none of these grouping schemes matched the phylogenetic results well, they provide a valuable indication that the germination type of zoospores might serve as a dependable morphological characteristic (Michetti et al. 2010)’.

Reviewer 3 Report
Comments and Suggestions for Authors
This manuscript under review establishes a new genus, Pseudostigeoclonium, within the Chaetophorales. I can follow the authors well, the present work is easy to read and quite interesting, nevertheless, there are some inconsistencies that need to be revised before publication:
In the Material and Methods part (line 350), the authors say that jModelTest was used to select the best-fitting evolutionary models for EACH marker! The models (plural) found (see the Results part) are relevant for the Bayesian analysis (with MrBayes), as well as for the maximum likelihood analysis (with RAxML). But the models are only given for the Bayesian method in the Results part (i.e., “for the BI analysis” (line 167). In addition, the authors write in the Material and Methods part (line 354) that (without further explanation) MrBayes and RAxML were used for the tree calculation, the latter with GTRGAMMA. It is therefore completely unclear whether the different models found for the different markers were used at all. Was the data set partitioned accordingly? Or was a model test carried out, but then (with regard to the model) a tree was calculated just with the default settings - that's what it looks like at the moment - the authors really need to clarify things here. How the models found by Modeltest have been used in the tree reconstruction (MrBayes and RAxML)?
Further, because of the manually refined alignment, it is necessary to provide the alignment as supplementary material. Otherwise, the present work would not be repeatable!
The figure 4 is much too small. Either redesign it or at least enlarge the lower area to landscape format. At the moment figure 4 is barely legible after a printout!
Author Response
Reviewer: 3
- This manuscript under review establishes a new genus, Pseudostigeoclonium, within the Chaetophorales. I can follow the authors well, the present work is easy to read and quite interesting, nevertheless, there are some inconsistencies that need to be revised before publication:
Re: Thank you for your comments. We have followed your suggestions and fixed our MS.
- In the Material and Methods part (line 350), the authors say that jModelTest was used to select the best-fitting evolutionary models for each marker! The models (plural) found (see the Results part) are relevant for the Bayesian analysis (with MrBayes), as well as for the maximum likelihood analysis (with RAxML). But the models are only given for the Bayesian method in the Results part (i.e., “for the BI analysis” (line 167). In addition, the authors write in the Material and Methods part (line 354) that (without further explanation) MrBayes and RAxML were used for the tree calculation, the latter with GTRGAMMA. It is therefore completely unclear whether the different models found for the different markers were used at all. Was the data set partitioned accordingly? Or was a model test carried out, but then (with regard to the model) a tree was calculated just with the default settings - that's what it looks like at the moment - the authors really need to clarify things here. How the models found by Modeltest have been used in the tree reconstruction (MrBayes and RAxML)?
Re: Thank you for your comments. Sorry for such unclear expressions.
Whether was Bayesian analysis or maximum likelihood analysis, the data set was partitioned accordingly. For the Bayesian analysis, different models found for the different markers were used at all. But for the maximum likelihood analysis, the same GTRGAMMA model was used for different markers. This was determined by different software parameters. I don't know if I explained it clearly or if you are satisfied. Thank you.
- Further, because of the manually refined alignment, it is necessary to provide the alignment as supplementary material. Otherwise, the present work would not be repeatable!
Re: Thank you for your comments. We highly agree with you. We have provided the alignment as supplementary material. Please see the file Appendix A. Data matrixes of the concatenated marker (18S rDNA + ITS2 + tufA) used for the ML and Bayesian phylogenetic tree reconstructions in Figure 4
- The figure 4 is much too small. Either redesign it or at least enlarge the lower area to landscape format. At the moment figure 4 is barely legible after a printout!
Re: Thank you for your comments. Sorry for the unclear display of the Figures. We have tried our best to modify it and hope to get your approval. Please see the Figure 4.
Round 2
Reviewer 1 Report
Comments and Suggestions for Authors
The authors ahve accepted all the comments and corrected properly.
Author Response
Reviewer: 1
- The authors have accepted all the comments and corrected properly.
Re: Thank you for your affirmation and comments.